# Uric Acid Causes Pancreatic β Cell Death and Dysfunction via Modulating CHOP-Mediated Endoplasmic Reticulum Stress Pathways

**DOI:** 10.3390/diseases13070213

**Published:** 2025-07-07

**Authors:** Xueyan Li, Yunan Chen, Lei Su, Jialin He

**Affiliations:** 1The Key Laboratory of Environmental Pollution Monitoring and Disease Control, Department of Nutrition, School of Public Health of Guizhou Medical University, Guiyang 561000, China; 2Guangdong Provincial Key Laboratory of Food, Nutrition and Health, Department of Nutrition, School of Public Health, Sun Yat-sen University (Northern Campus), Guangzhou 510000, China

**Keywords:** uric acid, β cell death, pancreatic β cell dysfunction, type 2 diabetes

## Abstract

Background: Uric acid has been proposed as a diabetogenic factor while its effect on pancreatic β cell function remains elusive. This study aimed to explore the impact of uric acid levels on β cell function and delineate its underlying molecular mechanisms. Methods: Both in vivo hyperuricemia diet-induced mouse models and in vitro pancreatic β cell models were utilized. Results: A progressive decrease in glucose-stimulated insulin secretion and increase in β cell apoptosis were observed in the hyperuricemia diet-induced mouse model, and these could be effectively restored by urate-lowering therapy. The dose- and time-dependent direct effects of uric acid on β cell apoptosis and insulin secretion were further confirmed in both INS-1E cells and primary isolated islets. Mechanistically, the primary role of expression of the endoplasmic reticulum stress marker C/EBP homologous protein (CHOP) was detected by RNA sequencing, and the inflammatory factor NLRP3 and pro-apoptotic genes were significantly upregulated by uric acid treatment. Conclusions: Together, our findings indicate a direct crosstalk between uric acid and β cells via CHOP/NLRP3 pathway, providing a new understanding of the diabetogenic effect of uric acid.

## 1. Introduction

Diabetes is a metabolic disease caused by a variety of causes and accompanied by various complications, which seriously endangers people’s health. At present, the prevalence of diabetes worldwide has brought serious threats and challenges to public health. In 2021, there were 529 million people with diabetes worldwide, and this number is expected to increase to 1.31 billion people by 2050 S [1]. Insulin resistance and pancreatic β cell function have been well recognized as two hallmarks of type 2 diabetes. Accumulating recent evidence has strengthened the decisive role of pancreatic β cell dysfunction in the development from the insulin resistance state or prediabetes state to the onset of overt diabetes [2,3,4,5]. In addition to nutritional interventions, targeted management of risk factors has emerged as a pivotal strategy in diabetes prevention and control.

Uric acid (UA), the end product of purine metabolism, is primarily synthesized in the liver and excreted by the kidneys. During this process, renal tubules reabsorb uric acid into the systemic circulation, and excessive purine or disorders in uric acid metabolism can also increase the level of uric acid in the blood; with the assistance of certain urate transporters, it may further enter other cells. Uric acid has been proposed as a potential diabetogenic factor for decades [6]. With the increased intake of high-energy and high-purine foods, the prevalence of hyperuricemia has risen dramatically in recent decades. Ample evidence from epidemiological investigations suggests that elevated serum uric acid (SUA) level is an independent risk factor for the development of type 2 diabetes [7,8], and for each increase of 1 mg/dL of SUA, the risk of type 2 diabetes increases by 17% [9]. Epidemiological evidence from large prospective cohort studies strongly supports the association between higher serum uric acid levels and the risk of diabetes [7,8], while the underlying molecular mechanisms by which uric acid induces diabetes remain unclear. Published epidemiological and experimental studies on the link between high uric acid levels and diabetes have mostly focused on its effects on peripheral insulin resistance. In a recent cross-sectional study of 6457 individuals, high serum uric acid levels were significantly correlated with insulin resistance indicators such as TyG and TyG-BMI [10], and other studies have shown consistent results; thus, the association of uric acid with insulin resistance has been supported in multiple studies [10,11,12]. In laboratory studies, it has been demonstrated that elevated uric acid levels interfere with insulin signaling pathways and decrease the bioavailability of endothelial nitric oxide (NO), thereby contributing to the onset of endothelial insulin resistance [13]. Additionally, in male KK-Ay/Ta mice, a type 2 diabetes model, increased serum uric acid levels were observed, which were positively correlated with blood glucose levels and HOMA-IR indices [14].

Limited studies have focused on the effect of uric acid on pancreatic β cell function [15,16,17]. A rare experimental study on an established hyperuricemia mouse model with uricase gene knockout showed increased apoptosis and decreased beta-cell function [18]; however, the mechanism by which high uric acid affects islet function is not clear. Thus, the effect of serum uric acid levels on pancreatic β cell function needs to be elucidated and the potential molecular mechanism of how uric acid acts on pancreatic β cells needs to be deciphered.

In the present study, we established a diet-induced hyperuricemia mouse model to investigate the dynamic change in pancreatic β cell function and further verified the direct role of uric acid on pancreatic β cell function in both β cell line and primary isolated islets. RNA-sequencing (RNA-seq) was applied to screen the differentially expressed genes induced by UA, and the critical pathways were further verified by Western blot (WB) and small interfering RNA (siRNA) analysis.

## 2. Materials and Methods

### 2.1. Animal Models

Male C57BL/6J mice were purchased from the Beijing Vitalstar Biotechnology (Beijing, China) and quarantined for 14 days before the investigation (8 weeks of age). All of the mice were maintained in a 12 h light/dark cycle at controlled room temperature and given free access to food and water. The mice were randomly allocated into three groups and were fed one of the following: standard chow diet (SCD), hyperuricemia-inducing diet (HUA), or HUA combined with allopurinol (ALL) (120 mg/L) in drinking water. The HUA consisted of 3% uric acid, 2% oxonic acid, and 95% SCD. All animal experiments were approved by the Animal Care and Utilization Committee of Sun Yat-sen University (Ethics Approval Number: NO.2022-010).

### 2.2. Intraperitoneal Glucose Tolerance Test (IPGTT) and Intraperitoneal Insulin Tolerance Test (IPITT)

Metabolic testing was conducted at different experimental time points following standardized protocols. For IPGTT measurements, after a 12 h fasting period, mice were administered 50% glucose solution (2 mg/g BW) via intraperitoneal injection as previously described [11]. IPITT evaluations required 4 h of fasting prior to intraperitoneal delivery of insulin (1 U/kg). Subsequently, the blood glucose levels in the IPGTT and IPITT were measured at 0, 30, 60, 90, and 120 min. The areas under the curve (AUC) of IPGTT and IPITT were calculated by GraphPad Prism 8.

### 2.3. Pancreas Immunostaining and β Cell Histomorphometry

Harvested mouse pancreatic tissues from mice were weighed and fixed in 10% neutral buffered formalin (Sigma-Aldrich, St. Louis, MO, USA), followed by paraffin embedding. For insulin detection, 4 μm sections were incubated with primary antibody (Anti-Insulin, Abcam, Waltham, MA, USA, #ab181547, RRID: AB_306130; 1:1000 dilution) at 4 °C for 16 h and hematoxylin staining. β cell mass was then calculated as previously described [19], and the distribution of islet size was quantified as the proportion of number of islets with area < 500 um^2^. β cell apoptosis in pancreatic islets was assessed through co-staining with insulin antibodies and TUNEL assay reagents (Promega, Indianapolis, IN, USA). Following incubation with Alexa Fluor 594-conjugated secondary antibodies (Life Technologies, Carlsbad, CA, USA), the percentage of apoptotic β cells was calculated as the ratio of TUNEL-positive nuclei to total insulin-positive cells. High-resolution imaging was performed using a Leica TCS SP5 II laser scanning confocal microscope.

### 2.4. Isolation, Culture, and Dissociation of Mouse Pancreatic Islets

Pancreatic islets were harvested from mice employing an established collagenase-based isolation method [20]. Specifically, islet extraction was performed using Collagenase P (Roche Applied Science, Mannheim, Germany); for pancreatic digestion, collagenase was perfused into the common bile duct. An amount of 2–3 mL of pre-cooled collagenase solution was slowly perfused and digested by shaking in a 37 °C water bath. A quantity of 50× *g* was centrifuged for 2 min at 4 °C and the supernatant was discarded. The solution was then resuspended and precipitated in 3 mL HBSS, and 800× g was centrifuged for 20 min. Following isolation, the islets were maintained in RPMI 1640 medium (supplemented with 10% fetal bovine serum) under standard culture conditions.

### 2.5. Cell Culture and Inflammatory Factors Assay

INS-1E β cells were cultured in DMEM containing 25 mM glucose supplemented with 10% fetal bovine serum, 5 μL/L β-mercaptoethanol, 2 mM L-glutamine, and antibiotics (100 IU/mL penicillin, 100 μg/mL streptomycin) at 37 °C with 5% CO_2_. INS-1E cells were treated with 0, 250, 500, 750 μmol/L uric acid for 24 h. The uric acid powder was purchased from sigma and prepared as a stock solution with a concentration of 100 mol/L; then, it was added to the cell for intervention, eventually reaching the required intervention concentration. Following the 24 h treatment period, culture supernatants were harvested for cytokine quantification. Levels of proinflammatory mediators (IL-1β and TNF-α) were determined using commercial ELISA kits (NeoBioscience, Shenzhen, China) according to the manufacturer’s protocols.

### 2.6. Glucose-Induced Insulin Secretion (GSIS) Assay

Primary islets and INS-1E cells were cultured using established protocols [20]. For GSIS evaluation, both cell types underwent sequential incubation in KRB buffer (0.1% fatty acid-free BSA) for 30 min, followed by exposure to either basal (3 mM) or stimulatory (25 mM) glucose conditions. For primary islets, dynamic insulin secretion profiles were obtained using a perfusion system as previously described [21], while eluted fractions were collected at indicated time points with the first phase defined as 0–10 min. Insulin concentration of the supernatant was measured by the commercial ELISA kit (80-INSMSU-E10, Mouse Insulin Ultrasensitive ELISA Kit, ALPCO Diagnostics, Salem, NH, USA, RRID:AB_2792981; 10-1250-01, Rat Insulin ELISA Kit, Mercodia, Uppsala, Sweden, RRID:AB_2811229).

### 2.7. Assessment of Cell Apoptosis

INS-1E cells were cultured as previously described. Quantities of 10 μmol/L PI working solution and 0.5 μg/mL DAPI working solution were mixed well and stored in the dark, after discarding the cell culture medium. The above-configured PI/DAPI staining solution was added, and this was placed in a cell incubator for 30 min, observing the cells at wavelengths of 488 nm and 545 nm. Cells were digested with 0.25% trypsin after the indicated intervention (0, 250, 500, 750 μmol/L for 24 h), and collected after being centrifuged at 300× *g* for 5 min and washed once with PBS; then, cells were gently resuspended and counted. A total of 1~5 × 105 resuspended cells were taken and then centrifuged at 300× *g* for 5 min, and 500 μL of diluted 1 × Annexin V Binding Buffer working solution was added to resuspend the cells. A quantity of 5 μL of Annexin V FITC and 5 μL of PI staining solution were added to the cell suspension, and this was analyzed with Flow Cytometer (BD Biosciences, San Jose, CA, USA).

### 2.8. Western Blot Analysis

Cellular and tissue lysates were prepared using RIPA buffer supplemented with a protease inhibitor cocktail (Roche, Tucson, AZ, USA). Protein quantification was performed via BCA assay prior to SDS-PAGE separation. Following electrophoretic transfer, membranes were probed with target-specific primary antibodies and corresponding secondary antibodies. Protein bands were finally detected using an ECL chemiluminescence system (Thermo Fisher Scientific, Chicago, IL, USA) and the following primary antibodies: anti-CHOP (2895, Cell Signaling Technology, Shanghai, China, RRID: AB_208925; 1:1000); anti-p-Perk (3179, Cell Signaling Technology, RRID: AB_2095853); anti-Perk (20582-1, Proteintech, RRID: AB_10695760; 1:1000); anti-cleaved-caspase3 (9661s, Cell Signaling Technology, RRID: AB_2341188; 1:1000); anti-Total-caspase3 (9662s, Cell Signaling Technology, RRID: AB_331439; 1:1000); anti-BAX (2772, Cell Signaling Technology, RRID: AB_10695870; 1:1000); anti-BCL2 (3498, Cell Signaling Technology, RRID: AB_1903907; 1:1000); anti-NLRP3 (13158, Cell Signaling Technology, RRID: AB_2798134; 1:1000); and anti-Pro-caspase1 (ab179515, Abcam, RRID: AB_2884954; 1:1000).

### 2.9. Total RNA Extraction and qRT-PCR Quantification

Total RNA was extracted from INS-1E cells using Trizol reagent (Life Technologies, Ontario, Canada). Briefly, cells were lysed directly in Trizol, followed by phase separation with chloroform and RNA precipitation with isopropanol. The RNA pellet was washed with 75% ethanol, air-dried, and dissolved in RNase-free water. RNA concentration and purity were determined by spectrophotometry. Subsequent cDNA synthesis was performed with 0.5 μg RNA input using PrimeScript RT Master Mix (Takara, Tokyo, Japan). Quantitative PCR analysis was conducted on an Applied Biosystems Prism 7000 system with SYBR Green chemistry. All reactions were performed in triplicate technical replicates, with β-actin serving as the housekeeping gene for normalization. Primer sequences are provided in Appendix A.

### 2.10. RNA Sequencing

The INS-1E cells were seeded in 6-well plates, and after being intervened with 750 μmol/L uric acid for 24 h, total RNA was extracted with 3 replicates in each group. The samples were collected and further processed and analyzed by LC Bio (Hangzhou, Zhejiang, China). Briefly, total RNA was quantified and mRNA was purified by Dynabeads Oligo (dT) (Thermo Fisher, CA, USA). The mRNA fragments were then transcribed into cDNA using SuperScript II reverse transcriptase (Invitrogen, Shanghai, China). Finally, the processed samples were subjected to PCR amplification, and 2×150 bp pair-end sequencing (PE150) was performed on an Illumina Novaseq 6000 (LC-Bio Technology CO Ltd., China, Hangzhou, China) according to the protocol recommended by the supplier. Differentially expressed genes with fold change > 2 and statistical significance (*p* < 0.05) were selected, using OmicStudio v2.0 tool to make heat maps.

### 2.11. siRNA Transfection

INS-1E cells were transfected with CHOP siRNA (HANBIO, Shanghai, China) or control siRNA (HANBIO) at 80 nM using Lipofectamine RNA iMAX (Invitrogen) according to the manufacturer’s instructions. For transfection, cells reached 50–70% confluency, after which siRNA was diluted in serum-free medium and gently mixed. In a separate tube, Lipofectamine RNAiMAX was diluted in an equal volume of Opti-MEM at the recommended ratio (1 μL RNAiMAX: 100 μL Opti-MEM) and incubated at room temperature for 5 min. The diluted siRNA was then fixed with the RNAiMAX solution, and the siRNA-RNAiMAX complexes were uniformly added to the cells. The culture plate was gently swirled to ensure distribution of the transfection complexes. After 24 h of transfection, the cells were then treated with 750 µmol/L uric acid for 24 h.

### 2.12. Statistical Analysis

Data are presented as mean ± SEM. Statistical comparisons were performed using SPSS 22.0 (IBM, Chicago, IL, USA). Between-group differences were analyzed by Student’s *t*-test for two-group comparisons. *p* < 0.05 was considered to be statistically significant. The use of GenAI for superficial text editing (e.g., grammar, spelling, punctuation, and formatting) does not need to be declared.

## 3. Results

### 3.1. Dynamic Impaired Insulin Secretion and Decreased β Cell Mass over Time in Hyperuricemia-Inducing-Diet-Fed Mice

To evaluate the possible causal role of SUA and pancreatic β cell dysfunction, we established a mouse model of elevated SUA by feeding C57BL/6J mice with hyperuricemia-inducing diet (HUA) as previously reported [11]. The mice were evaluated at different time points (2 weeks, 4 weeks, 8 weeks: 2 w, 4 w, 8 w). As expected, HUA-fed mice manifested a steadily and significantly elevated serum uric acid level compared with standard chow diet (SCD)-fed mice (Appendix A). HUA-fed mice exhibited a dynamic increasing fasting glucose (Appendix A), impaired glucose tolerance, and decreased peripheral insulin sensitivity compared with SCD-fed controls (Appendix A). The glucose-induced insulin secretion significantly decreased after eight weeks of HUA diet (25% lower than CON) and was effectively relieved by uric-acid-lowering drug allopurinol (Figure 1A).

Compared with an SCD control group, a dynamic decrease in pancreatic β cell mass was observed by immunostaining in HUA-fed mice and was rescued by allopurinol treatment (Figure 1B,C), and the number of smaller islets in HUA mice was significantly increased (Figure 1D). Pancreatic β cells apoptosis by TUNEL staining increased over time, and there were significant changes at eight weeks. The apoptotic β cells in the HUA group increased significantly compared with SCD control group at eight weeks, which was relieved by allopurinol treatment (Figure 1E,F).

### 3.2. Uric Acid Directly Impaired β Cell Secretory Function and Survival In Vitro

To assess the direct effect of uric acid on GSIS, β cell lines (INS-1E) were treated with different doses of UA (0, 250, 500, 750 μmol/L). According to a previous report, the dose we used covers the range of circulating uric acid levels in type 2 diabetic subjects [4]. Uric acid dose-dependently suppressed GSIS (INS-1E) and stimulatory index (8%, 41%, 60% in 250, 500, 750 μmol/L UA treatment, respectively, Figure 2A). Meanwhile, UA reduced insulin secretion in a time-dependent manner in the β cell line (Figure 2B). In addition, a similar inhibition effect of GSIS was also observed in an experiment involving a low dose of uric acid (250 μmol/L) produced in primary isolated islets from C57BL/J mice (Figure 2C). According to the phase characteristics of islet secretion, the insulin secretion in different phases of primary islets was measured after uric acid stimulation, and the first-phase GSIS was inhibited by up to nearly 30% compared to control (Figure 2D). As β cell loss is a hallmark of injury of β cell function, we next determined whether uric acid impacted the survival of β cell by propidium iodide (PI) staining (Figure 2E,F) and flow cytometry (FITC-PI) (Figure 2G,H). We found that β cell apoptosis dose-dependently increased in response to uric acid, which was in line with the increase in β cell apoptosis in vivo. Taken together, these data provided strong evidence that uric acid could directly impair the pancreatic β cell function and survival in vitro and in vivo.

### 3.3. Exploration of the Possible Molecular Mechanisms in Mediating the Damage Effect of Uric Acid

To further clarify the molecular mechanisms underlying the inhibitory effect of uric acid on β cell survival, we first used qPCR and WB to detect the possible role of the known membrane UA receptors including glucose transporter 9 (GLUT9) and Toll-like receptor 4 (TLR4) [22,23]. However, the expression of GLUT9 and TLR4 was not significantly altered by UA treatment (Figure 3A,B).

### 3.4. UA Regulated β Cell Dysfunction by Activating CHOP/NLRP3 Pathway

In order to explore the molecular mechanism of islet β cell damage caused by uric acid, we measured the transcriptome profile of β cell line (INS-1E) by RNA-seq. Overall, 19,289 expressed genes were found, including 138 differentially expressed genes, with 108 upregulated genes and 30 downregulated genes. Notably, the expression of pro-apoptotic marker C/EBP homologous protein (CHOP), also known as Ddit3, a classical marker of ER stress [24], significantly increased after treatment with uric acid (Figure 4A). The critical role of the CHOP-related ER stress pathway in mediating UA-induced β cell dysfunction was further confirmed by qPCR and WB, and we found that UA dose-dependently upregulated the protein and mRNA expression of CHOP in INS-1E cells (Figure 4B,C). PKR-like ER kinase (Perk) and Phospho-eukaryotic Initiation factor 2 alpha (p-eIF2α), two majors upstream of the CHOP, were dose-dependently upregulated by UA treatment (Figure 4B).

ER stress has been reported to activate NLRP3 inflammasome and apoptosis [25]; the NLRP3 expression and downstream inflammatory factors IL-1β were assessed. As shown in Figure 4B,C, the mRNA and protein levels of NLRP3 were increased, and the IL-1β was significantly upregulated by UA treatment, but the TNF-α expression did not change (Figure 5A). Apoptosis-related genes BAX and cleaved-caspase3 were dose-dependently increased in UA-treated INS-1E cells; no significant change in BCL2 was observed and p-eIF2α was gradually increased (Figure 5B). Consistent with in vitro experiments, the protein expression of CHOP and NLRP3 gradually increased over time in HUA-fed mice (Figure 5C).

### 3.5. Verification of the Critical Role of CHOP-Mediated ER Stress in Mediating the Damage Role of Uric Acid

In order to confirm the important role of the CHOP pathway in UA-induced β cell dysfunction, we silenced CHOP by siRNA in INS-1E, which showed that silencing CHOP abrogated the decline of UA-induced insulin secretion (Figure 6A) and reduced the activation of NLRP3; meanwhile, the expression of the p-eIF2α gene related to the CHOP signaling pathway decreased, and relative apoptosis molecules such as pro-caspase1 in the siCHOP+UA group exerted attenuated expression compared with siNC+UA group (Figure 6B,C). Taken together, these results suggest a crucial role of CHOP in the activation of the NLRP3-apoptosis cascade and the inhibition of insulin secretion induced by UA.

## 4. Discussion

The causal role of elevated serum uric acid levels in the pathogenesis of diabetes remains inconclusive and the mechanism of how uric acid takes effect on the pancreatic β cells remains to be investigated. In the current study, we found a progressive impairment of pancreatic β cell secretory function and reduction in β cell mass in a HUA-diet-fed mouse model; the direct deleterious effect of UA on pancreatic β cell function was further confirmed in in vitro INS-1E and primary isolated islet models. At the molecular level, we reported for the first time that CHOP-mediated ER stress was the central mechanism of UA-induced pancreatic β cell dysfunction. Our findings provide novel insights into the mechanism and pathogenesis of type 2 diabetes.

The associations between serum uric acid and β cell function reported by published studies largely varied. Participants with different glycemia status were studied in each study (nondiabetic, type 2 diabetes, or recent diagnosed GDM, respectively). These differences made it difficult to compare the results of one study with those of another. Conflicting results may be due to the different glycemia status of the study participants enrolled in different studies [15,26,27]. In laboratory studies, there have been some reports on the relationship between uric acid and glucose metabolism. For example, Scott found that the blood glucose level of rabbits could be raised to hyperglycemia after intramuscularly injecting UA (1 g/kg), suggesting that UA may play a role in diabetes [28]. Another study showed that in the hyperglycemia rat model with uricase gene knockout, the blood glucose of rats increased, which led to decreased insulin/glucose ratio [29]. Although these suggest the effect of uric acid on glucose metabolism, there are few studies that focus on uric acid and β cell function, and the association of serum uric acid with pancreatic β cell function over different times has not been reported before. The reason why uric acid causes impaired secretion of pancreatic islet β cells is also unclear.

In the present study, we found that the diet-induced hyperuricemia mouse model revealed an increasing apoptosis of pancreatic β cells using TNUEL, which targets the core markers of apoptosis (DNA damage), ultimately decreasing β cell mass and insulin secretion with the progress of hyperuricemia. In previous studies, it was found that uric acid has dual roles in DNA damage; Shahbaz Ahmad reported the role of uric acid as an antioxidant in alleviating mercury (II)-induced cellular and DNA damage in human blood cells [30], and an earlier study also reported UA inhibits peroxynitrite-mediated DNA damage [31]. Nevertheless, the negative effects of uric acid have received increasing attention. UA aggravates myocardial ischemia reperfusion-induced activation of the NLRP3 inflammatory cascade and pyroptosis by promoting ROS generation, and leads to DNA damage and apoptosis of myocardial cells [32]. In our study, uric acid exhibited negative effects and induced apoptosis, which was a manifestation of DNA damage in pancreatic β cells. In addition, we employed uric acid intervention in vitro and analyzed the dose- and time-dependent effects of uric acid on β cell function, and we discovered the impairment of insulin secretion in β cells and increased cell death, suggesting the direct effect of uric acid on islet β cells. These results indicate the causal effect of uric acid on pancreatic β cell function, and it is suggested that the decrease in insulin secretion function caused by uric acid is closely related to the increase in apoptosis and the decrease in the quality of islet β cells.

Previous studies have demonstrated the effects of uric acid on various cell types and tissues, along with investigations into its potential cellular receptors. To date, the two most extensively characterized uric acid receptors identified are Toll-like receptor 4 (TLR4) and glucose transporter 9 (GLUT-9) [22,33]. Therefore, we detected the expression of these two receptors by qPCR and WB. Interestingly, neither of them displayed significant differences following the treatment of UA, which reminded us to look for other ways to explore the possible mechanism.

To further explore the intensive mechanism of UA-induced dysfunction of β cells, we employed RNA-seq to excavate the gene profile after uric acid treatment; according to the results of the sequencing, CHOP signaling displayed significant increase. CHOP signaling is a pivotal part of ER stress, and many studies have reported CHOP activation in β cell dysfunction and apoptosis [34]. Meanwhile, studies have reported uric acid could function as a direct stimulus to activate ER stress in cardiomyocytes and vascular endothelial cells [35,36]. Here, we found that UA could certainly activate the CHOP cascade, including Perk, eIF2α, which are important for upstream regulation of CHOP [37]. In addition to CHOP, we also found upregulated NLRP3 after UA treatment. Interestingly, perturbed NLRP3 has been reported to regulate insulin resistance [38] or cell survival by UA stimulus in multiple cells including renal tubular epithelial cells and cardiomyocytes [39,40]. A number of studies have reported the intimate interaction between ER stress and NLRP3 inflammasome activation, and mitochondria-associated ER membranes (MAMs) are found to facilitate NLRP3 inflammasome assembly [41]. The CHOP-NLRP3 cascade has been indicated to be associated with glomerular podocyte damage in diabetic kidney disease [42]. In the present study, endoplasmic reticulum stress not only induces NLRP3 activation, but also promotes the release of downstream inflammatory factors of NLRP3; the most important factor is IL-1β. Next, to verify the central role of CHOP in UA-induced pancreatic cell dysfunction, siRNA targeting CHOP was used, and we found that inhibition of CHOP could not only effectively restore the decrease in glucose-stimulated insulin secretion but could also recover the activation of NLRP3. The expression of the downstream apoptosis-related marker caspase-1 was also reduced. Our study provides the first evidence that endoplasmic reticulum (ER) stress may serve as the central mechanism underlying uric acid-induced pancreatic β cell death and dysfunction. The other possible pathways are still worth investigating in future studies to completely understand the damaging effect of UA on pancreatic beta cells.

## 5. Conclusions

In conclusion, we demonstrated that uric acid could directly lead to morphological damage and increased apoptosis of islet β cells through the activation of CHOP/NLRP3, leading to impaired β cell secretion function. Our study provides critical insights into how diet-induced hyperuricemia may disrupt glucose homeostasis, and it advances our understanding of uric acid’s role in type 2 diabetes pathogenesis and facilitates the discovery of new druggable targets for type 2 diabetes.

## Figures and Tables

**Figure 1 diseases-13-00213-f001:**
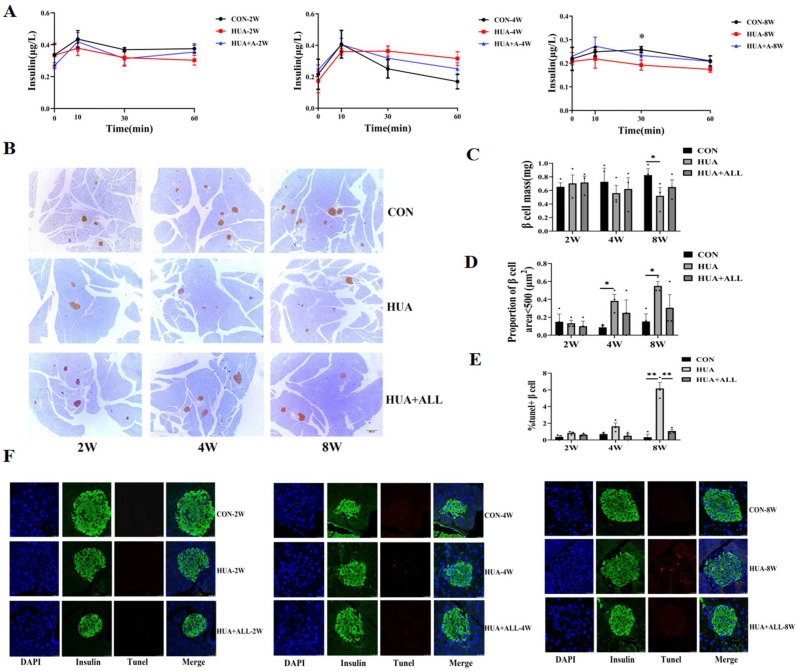
Uric acid affected islet function and β cell survival in vivo. (**A**) Mice were fed with HUA or CHOW diet with or without allopurinol (ALL) for different periods of time. Glucose-stimulated insulin secretion (GSIS) levels; * *p* < 0.05, ** *p* < 0.01, CON vs. HUA (*n* = 3 mice per group). (**B**) Insulin-positive cells determined by immunohistochemistry. (**C**) β cell mass. (**D**) Proportion of number of islets area <500 μm^2^ (*n* = 3 mice per group). (**E**,**F**) Representative images and statistical analysis of IHC staining of TUNEL (red) and insulin (green) in pancreases of mice (*n* = 3 mice per group). (The comparison between the two groups was conducted using Student’s *t*-test, * *p* < 0.05, ** *p* < 0.01).

**Figure 2 diseases-13-00213-f002:**
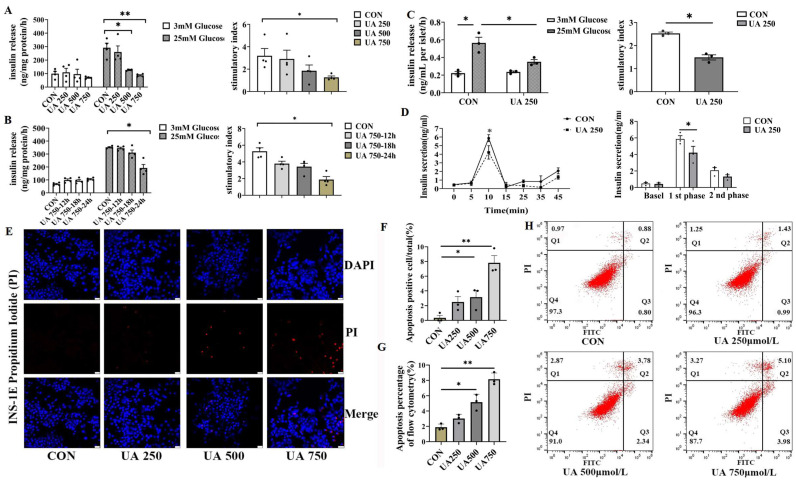
Uric acid directly impaired islet function and β cell survival in vitro. (**A**,**B**) Glucose-stimulated insulin secretion (GSIS) was assessed under basal (3.0 mM) and high-glucose (25.0 mM) conditions in INS-1E cells exposed to varying concentrations of uric acid (*n* = 4 mice per group); stimulatory index (calculated as the ratio of stimulated to basal insulin secretion) was used to evaluate β cell responsiveness. (**C**) GSIS levels and the corresponding stimulatory index of primary isolated islets from C57BL/6J mice after treatment with 250 μmol/L uric acid for 24 h (*n* = 3 mice per group). (**D**) Dynamic insulin secretion (first- and second-phase insulin) of the isolated islets from 12-week-old WT mice in response to 25.0 mM glucose stimulation using a perfusion system (*n* = 3 mice per group). (**E**) PI staining of apoptotic cells (scale bars, 50 μm) and (**F**) proportion of apoptosis-positive cells (*n* = 3 mice per group). (**G**,**H**) Apoptosis of INS-1E treated with indicated dose of uric acid for 24 h by flow cytometry (*n* = 3 mice per group). (The comparison between the two groups was conducted using Student’s *t*-test, * *p* < 0.05, ** *p* < 0.01).

**Figure 3 diseases-13-00213-f003:**
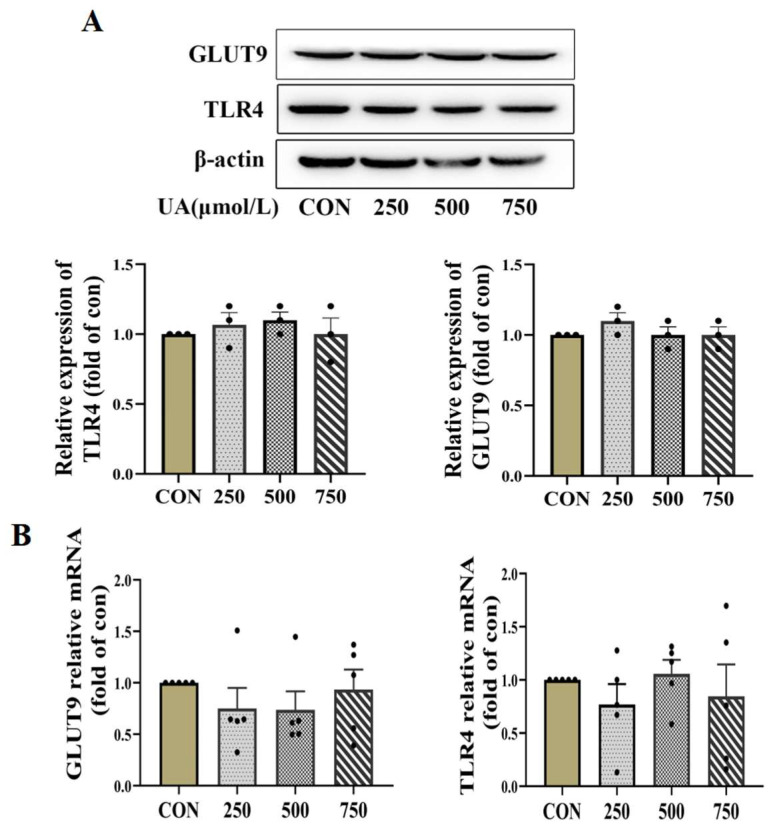
Uric acid mediator GLUT9 and TLR4 expressed in pancreatic β cells. (**A**,**B**) Protein (*n* = 3 mice per group) and mRNA expression (*n* = 5 mice per group) of GLUT9 and TLR4 in INS-1E cells treated with indicated concentration of uric acid for 24 h (the comparison between the two groups was conducted using Student’s *t*-test).

**Figure 4 diseases-13-00213-f004:**
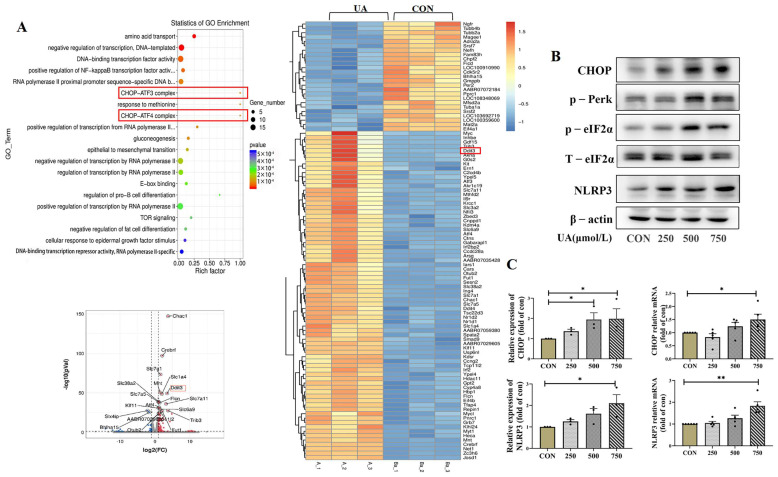
Transcriptome profile and CHOP pathway expression after uric acid treatment. (**A**) Genome (KEGG) enrichment plot, volcano map and heat map of INS-1E cells treated with 750 μmol/L uric acid for 24 h vs. control group (*n* = 3 mice per group). (**B**) Protein expression of CHOP, p-Perk, eIf2α, NLRP3 in INS-1E cells treated with indicated dose of uric acid for 24 h (*n* = 3 mice per group). The red box highlights the key gene CHOP, which usually interacts with ATF3 (Activating Transcription Factor 3) and ATF4 (Activating Transcription Factor 4) to jointly regulate the cellular stress response. (**C**) Protein and mRNA expression of CHOP and NLRP3 in INS-1E cells treated with indicated dose of uric acid (*n* = 3 mice per group). The comparison between the two groups was conducted using Student’s *t*-test, * *p* < 0.05, ** *p* < 0.01.

**Figure 5 diseases-13-00213-f005:**
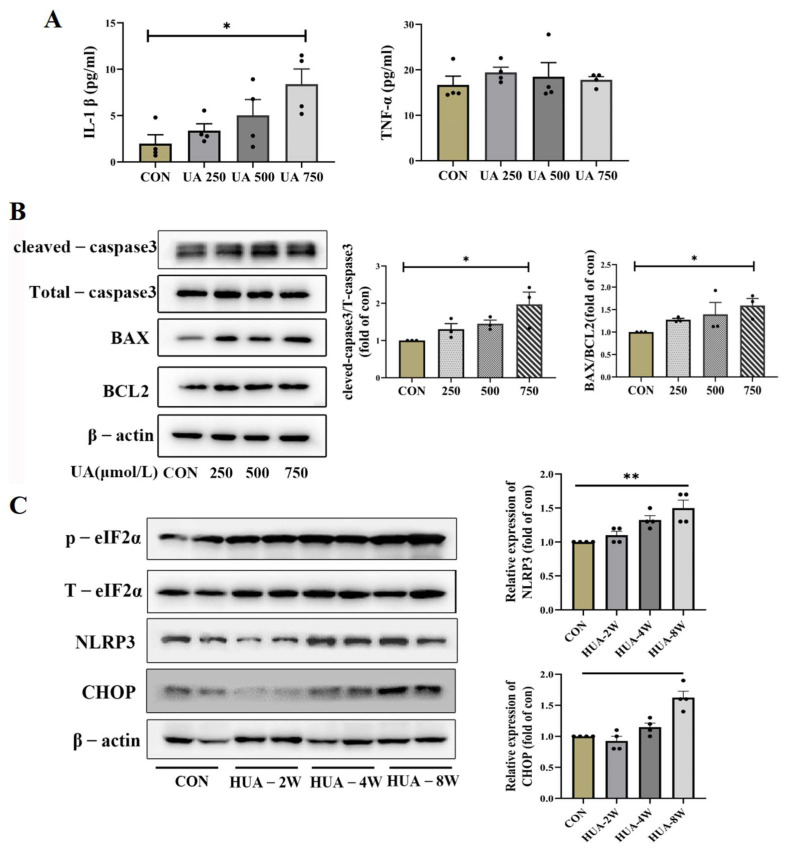
The concentration of inflammatory factor and expression of ER stress apoptosis. (**A**) The levels of IL-1β and TNF-α in supernatant of INS-1E cells after treatment with indicated dose of uric acid (*n* = 4 mice per group). (**B**) The expression of corresponding molecular apoptosis mechanism of CHOP signaling in INS-1E after treatment with 0, 250, 500, 750 μmol/L uric acid (*n* = 3 mice per group). (**C**) Protein expression of CHOP, NLRP3 and eIf2α in pancreas of HUA-fed mice (*n* = 4 mice per group) (* *p* < 0.05). The comparison between the two groups was conducted using Student’s *t*-test; * *p* < 0.05, ** *p* < 0.01.

**Figure 6 diseases-13-00213-f006:**
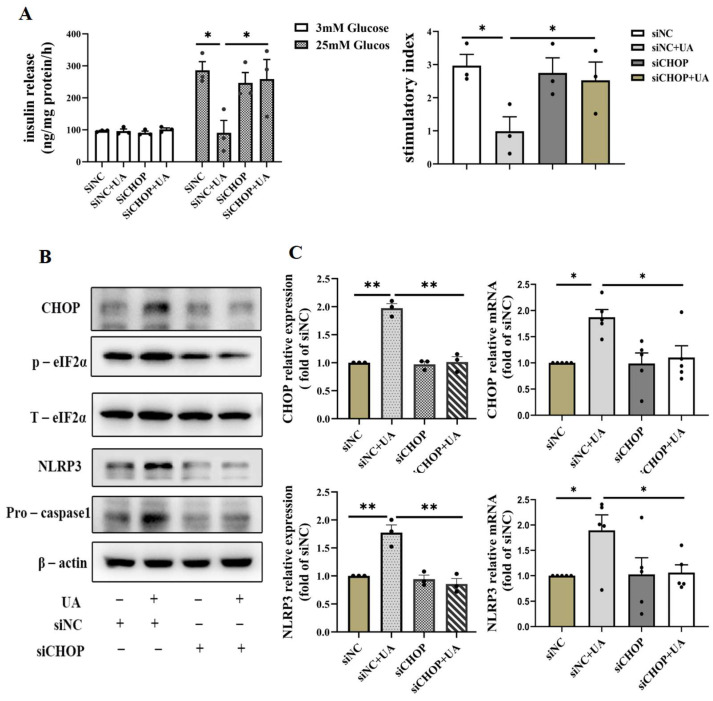
Effects of knockdown of CHOP on uric acid-induced β cell apoptosis and dysfunction. (**A**) GSIS and stimulatory index of INS-1E cells incubated with 750 μmol/L uric acid for 24 h after transfection with scrambled or CHOP targeting siRNA for 24 h (*n* = 3 mice per group). (**B**) The expression of NLRP3 and apoptosis relative gene of INS-1E cells incubated with 750 μmol/L uric acid for 24 h after transfection with scrambled or CHOP-targeting siRNA for 24 h (*n* = 3 mice per group). (**C**) The expression of CHOP protein ((*n* = 3 mice per group) and mRNA analysis (*n* = 5 mice per group) (* *p* < 0.05, ** *p* < 0.01). (The comparison between the two groups was conducted using Student’s *t*-test; * *p* < 0.05, ** *p* < 0.01).

## Data Availability

The original contributions presented in this study are included in the article. Further inquiries can be directed to the corresponding author.

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
