# Peer review of "Uric Acid Causes Pancreatic β Cell Death and Dysfunction via Modulating CHOP-Mediated Endoplasmic Reticulum Stress Pathways"

_diseases, 2025, doi:10.3390/diseases13070213_

Round 1
Reviewer 1 Report
Comments and Suggestions for Authors
This work is devoted to uric acid that causes pancreatic β cell death and dysfunction via modulating CHOP-mediated endoplasmic reticulum stress pathways. Here the impact of uric acid levels on β cell function were studied. Moreover, its underlying molecular mechanisms were delineated. The findings of this work indicate a direct crosstalk between UA and β cell via CHOP/NLRP3 pathway, providing a new understanding of the diabetogenic effect of UA. I think that this article may be published after minor revision.
Notes:
- Authors should avoid any abbreviations in the Abstract of the manuscript.
- Were the mice quarantined for 14 days after purchase from the Beijing Vitalstar Biotechnology (China) before the investigation?
- The insert drawings in the Figure 2 should be increased for clarity. It’s difficult to understand these drawings.
- Conclusions of this work should be added. Main findings of this work should be reflected in Conclusion more detailed.
Author Response
Dear Reviewer, thank you for your opinion. Your opinions are very helpful for improving the quality of my manuscript. Please refer to the attachment for my reply to your opinions.

Reviewer 2 Report
Comments and Suggestions for Authors
The progress of medicine has led not only to new therapies but also to more effective diagnostic tools. On the other side, the most promising factor in medical care is education. Currently, many different nutrition profiles have been promoted by the mass media without a scientific background—their uncritical application can cause biochemical and physiological cell dysfunction. Therefore, the article entitled “Uric acid causes pancreatic β cell death and dysfunction via modulating CHOP-mediated endoplasmic reticulum stress pathways” can be valuable.
The uric acid is one of the best extra and intracellular antioxidants, which is the central point of social interest. Therefore, the criticism of its influence on the Langerhans pancreas island is valuable. The pancreas cancer is the one of the most difficult to cure with a low good prognosis.
The methods described and applied for this study are correctly selected and described. The animal model and statistic analysis are correct.
Moreover, the article is well-written and readable, with correctly selected references.
I have three critical remarks:
- Authors should describe the source of uric acid in the cell
- The attention should be put on the source of protein in the human nutritional profile
- The influence of UA on DNA damage formation should be discussed (negative and protective role).
In conclusion, after answering my questions, I can recommend the article for publication.
Author Response

(The authors gave the same response as above.)

Reviewer 3 Report
Comments and Suggestions for Authors
The article by Xueyan Li et al is devoted to evaluate the role of uric acid (UA) in development of type II diabetes. In vivo hyperuricemia diet-induced (HUA) mouse model allowed the authors to elucidate decrease of glucose stimulated insulin secretion (GSIS) levels and β-cells mass and increased apoptosis of β-cells induced by UA. In vitro pancreatic β-cells model demonstrated time and dose-dependent effect of UA on apoptosis of β-cells and decrease of insulin secretion. The survival of β-cells was confirmed by Propidium iodude (PI) staining followed by flow-cytometry. It is important to note that deleterious effects of UA were almost completely restored by allopurinol treatment (urate-lowering therapy).
In search of molecular mechanisms of UA-induced disfunctions the authors for the first time have shown by qPCR and WB that the level of Glut9 and Tall-like receptor4 (TLR4), a known membrane UA receptors, is not changed. Instead the analysis of transcriptome profile of INS-1E cell line associated UA regulated β-cells disfunction with activation of CHOP/NLRP3 signaling pathway and ER-stress. The levels of CHOP and NLRF protein and mRNAs expression increased proportionally with increasing of UA-dose (0, 250, 500, 750)μM. Simultaneously the inflammatory factor IL-1β was upregulated by UA while expression of TNF-ά did not change. Also apoptosis related genes BAX and cleaved caspase3 were increased in a dose-dependent manner.
The article is interesting, contains many useful information and new observations and obviously should be published in Diseases. At the same time there are some comments and questions to the authors which are not of a fundamental nature, but would help improve the quality of the article.
- Tunel staining indicating apoptotic cells in Fig.1F (column 3) is almost not visible while PI staining used for the same purpose in Fig.2E is seen much better. Why not to use PI staining or both together as described in Materials and Methods?
- Can the authors explain why the area marked by insulin staining of HUA+all in Fig.1F is obviously smaller than HUA especially after 2 weeks and also after 8 weeks of observation as well? There are no comments about this phenomenon in Discussion.
- Fig.2D-F in Suppl. illustrates insulin tolerance test so the authors should include explanation of the method in Materials and Methods section.
Author Response

(The authors gave the same response as above.)
